# Policy-Engineering Optimization with Visual Representation and Separation-of-Duty Constraints in Attribute-Based Access Control

**Wei Sun *** **, Hui Su and Huacheng Xie**

Center of Network Information and Computing, Xinyang Normal University, Xinyang 464000, China;
suhuixy@xynu.edu.cn (H.S.); xiehc@xynu.edu.cn (H.X.)
**\*** Correspondence: sunny810715@xynu.edu.cn

**Abstract:** Recently, attribute-based access control (ABAC) has received increasingly more attention and has emerged as the desired access control mechanism for many organizations because of its flexibility and scalability for authorization management, as well as its security policies, such as separation-of-duty constraints and mutually exclusive constraints. Policy-engineering technology is an effective approach for the construction of ABAC systems. However, most conventional methods lack interpretability, and their constructing processes are complex. Furthermore, they do not consider the separation-of-duty constraints. To address these issues in ABAC, this paper proposes a novel method called policy engineering optimization with visual representation and separation of duty constraints (PEO_VR&SOD). First, to enhance interpretability while mining a minimal set of rules, we use the visual technique with Hamming distance to reduce the policy mining scale and present a policy mining algorithm. Second, to verify whether the separation of duty constraints can be satisfied in a constructed policy engineering system, we use the method of SAT-based model counting to reduce the constraints and construct mutually exclusive constraints to implicitly enforce the given separation of duty constraints. The experiments demonstrate the efficiency and effectiveness of the proposed method and show encouraging results.

**Keywords:** attribute-based access control; policy engineering; visual authorization representation; separation-of-duty constraints

## 1. Introduction

With the rapid development and comprehensive application of network information technology, there is a large amount of storage required and many exchanges in large-scale and complex information-management systems [1]. Organizations adopt access control mechanisms to ensure the system security, and the role-based access control (RBAC) mechanism has been the main standard for most organizations over the last three decades. However, there is only one role attribute in RBAC systems on which users and objects can depend. The RBAC mechanism is identity-dependent and lacks flexibility, particularly in large-scale collaborative environments. As an alternative, attribute-based access control (ABAC) has been developed recently. In ABAC, a request for accessing any resource is permitted or denied based on the attributes assigned to the requesting user, the attributes assigned to the requested object, the environment condition where the request is made, and an authorization policy [2]. An ABAC policy is a set of authorization rules that includes various combinations of attribute-value pairs of users, objects, and environments, as well as operating privileges. If a user makes a request to access an object, an authorization rule to satisfy the access request is sought. Its flexibility, scalability, and identity-less properties overcome the limitations of RBAC, and make ABAC very attractive for

use in collaborative systems like cloud computing and the internet of things [3]. For the successful implementations of ABAC mechanisms in commercial organizations, the identification of a suitable set of authorization rules and the construction of a good ABAC system are critical tasks. This process, known as policy engineering [4,5], is regarded as one of the most difficult and costliest components for implementing the ABAC mechanism. Similar to role engineering in RBAC, there are also two main approaches for constructing policy-engineering systems: top-down [6] and bottom-up [7–9]. For the former, rules are specified by precisely evaluating and splitting the business processes into smaller independent units that are then associated with access permissions. However, this approach can ignore the existing access modes in the organization and is also time-consuming, labor-intensive, and error prone. For the latter, rules are derived from existing access permissions, and the architectural structure of ABAC can be automatically constructed. The bottom-up approach, also called policy mining, has gained much interest and considerable popularity in the last few years.

The policy-mining problem in ABAC involves discovering a suitable ABAC policy from a traditional access control mode, such that the access authorizations covered by the policy are consistent with the traditional access permissions. Xu and Stoller [10] first proposed a well-known bottom-up mining approach (simply represented as Xu-Stoller) and derived an ABAC policy from access control lists and the corresponding attribute data. Das et al. [11] considered that the policy-engineering problem in ABAC and the role-engineering problem in RBAC are similar and equally important for the construction of the corresponding access control models and presented a detailed survey of the two techniques. Actually, to enhance the interpretability of policy mining, it is necessary to cluster users, objects, or environments with the same attribute properties, similar to the clustering of users or permissions in the role-mining problem. However, due to the diversity of the attribute properties of entities and the variability of accesses, the mining scale is large and complex using conventional policy-mining methods.

A key characteristic of ABAC is that it allows the specification and enforcement of various types of constraints, such as the separation-of-duty (SOD) constraints, cardinality constraints, binding-of-duty (BOD) constraints, and user-capability (UC) constraints, which are irrelevant to the access control mechanism implemented in the system and can reflect the different security requirements of organizations and ensure the security of ABAC systems [12]. As a significant security policy discussed in this paper, the SOD constraint prevents an individual from performing all the steps involved in an important task, as a single user is more likely to abuse his or her privileges, while multiple users can supervise with each other while performing a task. Typically, a $k$-$n$ SOD constraint requires that at least $k$ users complete a special task that requires $n$-many operating permissions [13] and has been widely used in the banking industry and in military systems. However, none of the conventional policy-engineering methods consider the SOD constraint. Moreover, although mutually exclusive authorization rules (MEAR) constraints have been used to enforce a given separation of authorization rules (SOAR) constraint in an existing ABAC system [13], a SOAR constraint takes the form of authorization rules as its input, while an SOD constraint takes the form of access tuples. Therefore, it is necessary to convert the access tuples into authorization rules to facilitate the enforcement of the SOD constraints, which is also an interesting issue.

To address the above issues, this paper proposes a novel method called policy-engineering optimization with visual representation and separation-of-duty constraints (PEO_VR&SOD). In summary, the main contributions of this work are as follows:

(1)	To reduce the mining scale and enhance the interpretability of policy mining, we use the visual technique with Hamming distance to rearrange, portray, and partition an original authorization matrix and discover a minimal set of authorization rules from rearranged submatrices. We present a policy mining algorithm and compare its performance to the existing methods.

(2)	To verify whether SOD constraints can be satisfied in a constructed policy engineering system, we convert the SOD constraints into SOAR constraints using the method of SAT-based

model counting. We construct MEAR constraints from the SOAR constraints to implicitly enforce the given SOD constraints and evaluate the performance of the PEO_VR&SOD.

The rest of the paper is organized as follows. We discuss the related work in Section 2 and present some preliminaries that are discussed in the following sections in Section 3. Section 4 proposes a novel method for policy engineering optimization, which involves two aspects: (1) policy mining with visual representation and (2) policy optimization with separation of duty constraints. We present experimental evaluations and compare their performance with existing studies in Section 5. Section 6 concludes the paper and discusses future work.

## 2. Related Work

### 2.1. Research on Policy Engineering in ABAC

Various methods have been proposed for ABAC policy engineering. Depending on whether minimizing the number of mining rules is considered an optimized objective, existing studies mainly fall into the following two categories: general policy engineering and optimized policy engineering.

To avoid the potential risks of permitting unauthorized accesses, Krautsevich et al. [14] presented a risk based ABAC policy engineering problem that assessed the potential risk for each possible access while minimizing the total risk in the ABAC system. Biswas et al. [15] proposed a label based ABAC model using the method of enumeration for the construction of ABAC policies, each of which included only one user attribute and one object attribute. Narouei et al. [6] proposed a top-down ABAC policy engineering framework using a deep recurrent neural network, which derived authorization rules from unrestricted natural language documents. Iyer et al. [16] presented a novel method for ABAC policy mining to construct positive and negative authorization rules. Das et al. [17] proposed a hybrid approach for policy engineering in ABAC, which first used a top-down approach and then used a bottom-up approach to construct authorization rules. Although existing approaches are capable of constructing ABAC policies, the number of attribute-value pairs in any constructed rule is also critical. The time needed for access decisions increases with an increasing number of attribute-value pairs included in any rule. To improve the efficiency of the mining process, Gautam et al. [18] regarded the number of attributes included in any rule as a weight and presented a constrained policy mining algorithm in ABAC that constructed a set of authorization rules from an access control matrix, such that the weight of each rule was less than a specified value, and the sum of the total weights of the rules was minimized.

To discover an optimal set of ABAC rules from conventional access modes, Das et al. [19] presented an ABAC policy mining algorithm that included environmental attributes that used the Gini impurity to form an ABAC policy while minimizing the number of rules. Talukdar et al. [20] showed that the policy-mining problem is equivalent to identifying a set of functional dependencies in relational databases. The authors first proposed an ABAC policy mining algorithm called ABAC-FDM. Although this algorithm identified all the potential rules, the complexity of the algorithm was exponential. As an alternative, the authors next proposed another more efficient mining algorithm called ABAC-SRM, which discovered a suitable and minimal set of rules from candidate rules. To obtain a minimal set of ABAC rules in multi-cloud collaborations, John et al. [21] defined a cross-domain rule-mining problem called CDRMP that was proven to be NP hard and provided a heuristic solution. They also defined a cross-domain rule-mining problem with access relaxations under dynamic collaborations [22] called $\beta$-CDRMP and DCDRMP and presented the heuristic solutions for mining minimal sets of authorization rules. Besides mining positive rules in the cross domain, John et al. [23] considered the mining of negative rules from a given set of multi-cloud access requests, thereby defining a novel problem called CDRMP-H, and presented a solution to further reduce the number of mining rules, including the positive and the negative.

## 2.2. Research on Constraints in ABAC

Various studies have focused on specifying constraints in ABAC systems. Jin et al. [24] proposed a unified model called ABAC$\alpha$ that could configure three classical models. They also presented a policy specification language that specified constraints on attribute-assignment relationships. Bijon et al. [25] proposed an attribute-based constraint specification language (ABCL) for specifying a variety of constraints. The ABCL was used to specify constraints on a single attribute or on multiple attributes of a particular entity. Although the constraints in ABAC$\alpha$ were event-dependent, the constraints in ABCL were uniformly enforced no matter which attribute assignment was changed. Helil et al. [26] first examined the potential relationships between subjects and objects and then proposed an attribute-based access control constraint based on subject similarity. Jha et al. [27] presented a specification and verification of the SoD constraints in ABAC systems, analyzed the complexity of enforcing the SOD constraints and proposed an approach for solving them. Roy et al. [12] presented an employee replacement problem (ERP) in ABAC, in which the SOD constraints, BOD constraints, and UC constraints were simultaneously taken into consideration. The authors also provided a solution for verifying whether a particular subset of users in an ABAC system could be replaced with a smaller set of users while satisfying different types of constraints. Additionally, Alohaly et al. [28] proposed an automated framework to extract the ABAC constraints from natural language policies.

## 2.3. Research on Visual Representation for Access Information

Several techniques have been proposed for the visual representation of access information in a matrix form. To reduce the complexity of role-mining problems, Colantonio et al. [29] divided the user-permission-assignment dataset into several subsets and proposed a visual method for role mining. To reduce the mining scale, Verde et al. [30] converted role mining into a clustering problem, which compressed the division into a single sample, visually extracted similar features from multiple divisions, and ensured the integrity of the mining results. To facilitate the visual elucidation of access control matrices, Das et al. [31] introduced a novel method called visual mining of ABAC polices (VisMAP), which derived rules from the visual representation for a given authorization matrix while minimizing the number of rules. To enhance interpretability of the role-mining process, a novel method for role optimization is proposed in our work [1], which uses partitioning and compressing technologies to validate the accuracy of the method. Zheng et al. [32] provided a concrete example to visually specify ABAC rules and proposed a novel approach for detecting conflicts by transforming rules into a set of binary sequences.

## 2.4. Characteristics of Our Work

Two main limitations are apparent in the existing studies. The first limitation is that the policy mining scale is very large, and the mining process itself is confusing, complex, and lacks interpretability. The second limitation is that most existing policy engineering methods do not consider the SOD constraints, though various studies on different types of constraints in ABAC have been developed and assume that ABAC systems already exist in advance. However, in many cases, the systems are completely unknown and need to be constructed. Hence, in this work, we propose a novel policy engineering method (PEO_VR&SOD) with two main characteristics: (1) Visual representation technique with Hamming distance is used to reduce the policy-mining scale and enhance interpretability, and (2) the enforcement of SOD constraints is taken into consideration in the policy optimization. We also compare the performance of the PEO_VR&SOD with that of the existing methods through experiments.

## 3. Preliminaries

### 3.1. Basic Components of ABAC

According to the NIST standard for ABAC systems [33], we present the basic components of ABAC as follows:

(1) $U$ represents a finite set of requesting users. Each element of the set is denoted as $u_i$, where $1 \leq i \leq |U|$.

(2) $O$ represents a finite set of requested objects. Each element of the set is denoted as $o_i$, where $1 \leq i \leq |O|$.

(3) $OP$ represents a finite set of the operations allowed to be performed on the objects in an ABAC system. Each element of the set is denoted as $op_i$, where $1 \leq i \leq |OP|$. For instance, if there are only two operations allowed in a system: *read* and *write*, then we represent OP = {*read*, *write*}.

(4) $E$ represents a finite set of environments in which authorizations are made, such as time and locations. These authorizations are independent of users and objects. Each element of the set is denoted as $e_i$, where $1 \leq i \leq |E|$.

(5) $UA$ represents a finite set of attribute names of users. Each element of the set is denoted as $ua_i$, where $1 \leq i \leq |UA|$. User attribute $ua_i$ can associate several values. If we use $Val_{ua_i}$ to represent the one-to-many mapping of $ua_i$ onto a set of attribute values, it can be formalized as:

$$\forall ua_i \in UA : Val_{ua_i} = \left\{ val_{ij}^u | 1 \leq j \leq n_i^u, n_i^u \in Z^+ \right\} \cup \{null\}, \tag{1}$$

where *null* indicates that the corresponding attribute values of the user is unknown or uncertain. For instance, user attribute *Role* in a hospital can take values of Doctor, Nurse, and Patient, and then we represent $Val_{Role}$ = {Doctor, Nurse, Patient}.

(6) $UAV$ represents a finite set of all the possible attribute name–value pairs of users. Each element of the set is denoted in the form of the equality $ua_i = x_i$, where $ua_i \in UA$, $x_i \in Val_{ua_i}$. For instance, if there are two attributes of users: *Role* and *Specialty*, where *Role* can take values of Doctor, Nurse and Patient, and *Specialty* can take values of Cardiology, Medicine, and Pediatrics, then $UAV$ is represented as: {*Role* = Doctor, *Role* = Nurse, *Role* = Patient, *Specialty* = Cardiology, *Specialty* = Medicine, *Specialty* = Pediatrics}.

(7) $UUAV \subseteq U \times UAV$ represents a many-to-many assignment of users to their attribute name-value pairs. It can be formalized as:

$$UUAV = \{(u_k : ua_1 = x_1, ua_2 = x_2, \ldots, ua_i = x_i, \ldots) | u_k \in U, (ua_i = x_i) \in UAV\}. \tag{2}$$

(8) $OA$ represents a finite set of attribute names of objects. Each element of the set is denoted as $oa_i$, where $1 \leq i \leq |OA|$. The object attribute $oa_i$ is also associated with several values. If we use $Val_{oa_i}$ to represent the one-to-many mapping of $oa_i$ onto a set of attribute values, it can be formalized as:

$$\forall oa_i \in OA : Val_{oa_i} = \left\{ val_{ij}^o | 1 \leq j \leq n_i^o, n_i^o \in Z^+ \right\} \cup \{null\} \tag{3}$$

where *null* indicates that the corresponding attribute values of the object is unknown or uncertain. For instance, object attribute *Department* in a hospital can take values of Cardiology, Dermatology, and Gynecology, and then we represent $Val_{Department}$ = {Cardiology, Dermatology, Gynecology}.

(9) $OAV$ represents a finite set of all the possible attribute name-value pairs of objects. Each element of the set is denoted in the form of the equality $oa_i = y_i$, where $oa_i \in OA$, $y_i \in Val_{oa_i}$. For instance, if there are two attributes of objects: *Department* and *RecordOf*, where *Department* can take values of Cardiology, Dermatology, and Gynecology, and *RecordOf* can take values of Doctor, Nurse, Patient, and Staff, then $OAV$ is represented as: {*Department* = Cardiology, *Department* = Dermatology, *Department* = Gynecology, *RecordOf* = Doctor, *RecordOf* = Nurse, *RecordOf* = Patient, *RecordOf* = Staff}.

(10) $OOAV \subseteq O \times OAV$ represents a many-to-many assignment of objects to their attribute name–value pairs. It can be formalized as:

$$OOAV = \{(o_k : oa_1 = y_1, oa_2 = y_2, \ldots, oa_i = y_i, \ldots) | o_k \in O, (oa_i = y_i) \in OAV\}. \tag{4}$$

For the sake of brevity, we assume that environments are irrelevant to policy engineering in an ABAC system and thus do not consider the environmental factors in this paper.

### 3.2. Basic Policy-Mining Problem in ABAC

Besides the basic components of ABAC, the other components involved in traditional ABAC policy mining [20] can be presented as follows:

(1)   *A* represents a set of all possible authorizations that occur in an ABAC system. Each element of the set is represented as $a = <u,o,op>$, which allows user $u$ to perform operation $op$ on object $o$, where $u \in U$, $o \in O$, $op \in OP$.

(2)   *P* represents an ABAC policy, which is also referred to as a set of authorization rules *AR*. Each element *ar* in *AR* is denoted in a 3-tuple form $<UAV',OAV',OP'>$, where $UAV' \subseteq UAV$, $OAV' \subseteq OAV$, $OP' \subseteq OP$.

For the sake of simplicity, assume that any rule *ar* comprises only one operation, and *ar* is simply represented as $<UAV', OAV', op>$. Figure 1 presents the factors influencing the access decisions in ABAC, where single arrow heads represent which factors influence the access decisions and double arrow heads represent the one-to-many or many-to-many mappings between these influencing factors and other components of ABAC.

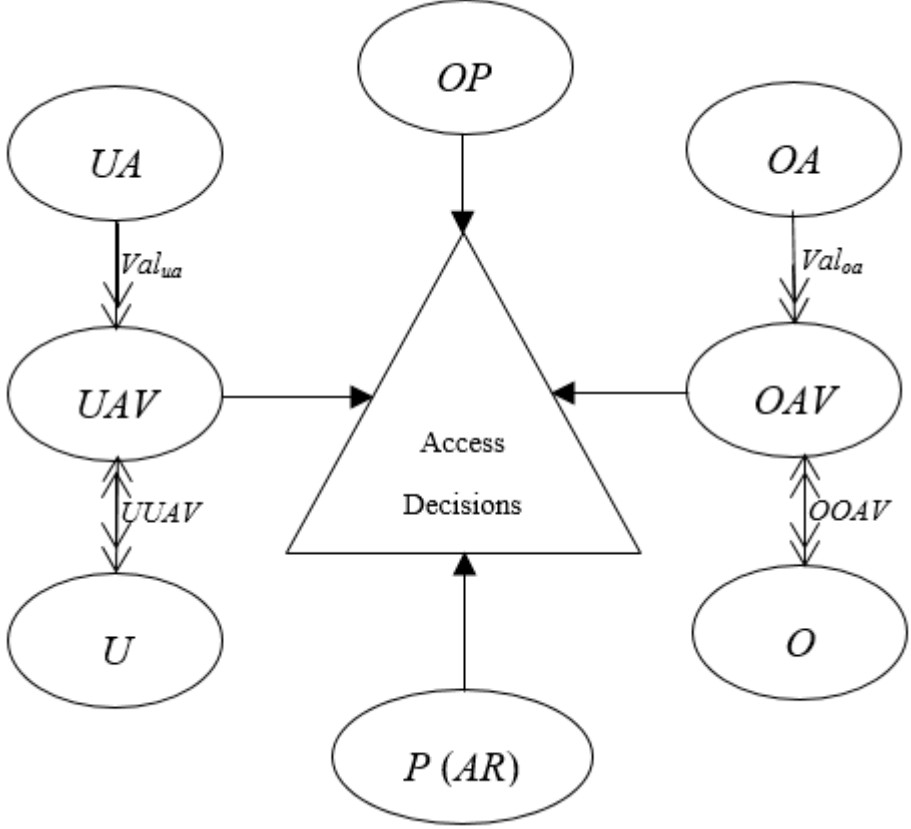

**Figure 1.** Factors influencing access decisions.

The basic ABAC policy-mining problem [21–23], in terms of the above elements, states that, given a set of authorizations $A = \{a_1, a_2, \dots\}$, a mapping list *UUAV* and a mapping list *OOAV* must find a set *AR* of authorization rules that can cover all the authorizations in *A*. Specifically, for any authorization $a = <u,o,op>$, user $u$ can perform operation $op$ on object $o$ if and only if some combination of several attribute–value pairs of $u$ in the *UUAV*, as well as those of $o$ in the *OOAV*, can match a rule with $op$ in *AR*. Furthermore, the number of mining rules is minimized.

### 3.3. Enforcement of SOD Constraints in ABAC

The SOD constraint includes static SOD and dynamic SOD. In this paper, we only consider the former, as we do not deal with environmental attributes, such as time and locations, and instead simply such attributes as SOD. The *k-n* SOD constraint [13] is expressed as $sod < \{t_1, t_2, \ldots, t_i \ldots, t_n\}, k >$, where $n$ and $k$ are integers, such that $2 \leq k \leq n$. Each $t_i$ is an access permission that is represented as a 2-tuple form $(op, o)$, where $op \in OP$, $o \in O$. We represent the set of such *sod* constraints as $\omega = \{sod_1, sod_2, \ldots\}$.

To cover all the access tuples in a *k-n* SOD constraint, the authorization rules can be used as a substitute for the tuple. Similar to the *k-n* SOD constraint, the *k-n* SOAR constraint [13] is expressed as $soar < \{ar_1, ar_2, \ldots ar_n\}, k>$, where each $ar_i$ is an authorization rule, and $n$ and $k$ are integers, such that $2 \leq k \leq n$. We represent the set of such *soar* constraints as $\xi = \{soar_1, soar_2, \ldots\}$.

MEAR constraints can be used to enforce SOAR constraints. The *t-m* MEAR constraint $mear < \{ar_1, ar_2, \ldots ar_m\}, t >$ [13] conveys that, for the given $m$ rules $ar_1, ar_2, \ldots ar_m$ in an ABAC system, no user is allowed to have $t$ or more of these $m$ rules, where each $ar_i$ is an authorization rule, and $m$ and $t$ are integers, such that $2 \leq t \leq m$. We represent the set of such *mear* constraints as $\psi = \{mear_1, mear_2, \ldots\}$.

### 3.4. Hamming Distance

Since the authorization list for some operations is a Boolean matrix, each row (or each column) can be regarded as a binary vector of the same length. The well-known Hamming distance [34], which is widely used to measure the distance between two different equal-length vectors, can identify clusters of the same (or similar) use–object pairs. Hamming distance states that, given two equal-length Boolean vectors $x$ and $y$, the Hamming distance between x and y, denoted as $Dis(x, y)$, is the number of positions where the vectors take different values for the same column position.

For instance, given two row vectors, $x =$ "100110" and $y =$ "110011", $Dis(x, y) = 3$. Clearly, the distance between any two rows in $A_{op}$ increases with an increasing number of column positions that take different values.

### 3.5. SAT-Based Model Counting

The well-known SAT solver [35,36] is commonly used to solve the model counting problem. An instance of the SAT-based model counting problem is a Boolean formula made up of different clauses. Each clause is a disjunctive or conjunctive form of Boolean variables. A Boolean formula, which can be expressed using the conjunction or disjunction of different clauses, is categorized into the conjunctive normal form (CNF) and the disjunctive normal form (DNF). Clearly, there are different truth assignments for a Boolean formula. Essentially, the model counting problem is to find truth assignments for a specific Boolean formula.

## 4. Proposed Method

In this section, we propose a novel method called PEO_VR&SOD, which includes two aspects: (1) ABAC policy mining with visual representation and (2) policy optimization with separation-of-duty constraints. Its flow chart is presented in Figure 2.

As shown in the figure, to reduce the mining scale and enhance the interpretability of policy mining, we adopt the visual technique with Hamming distance to rearrange, portray and partition an original authorization matrix, and discover a minimal set of authorization rules from rearranged submatrices. Subsequently, we utilize the method of SAT-based model counting to verify whether the separation of duty constraints can be satisfied in the constructed ABAC system.

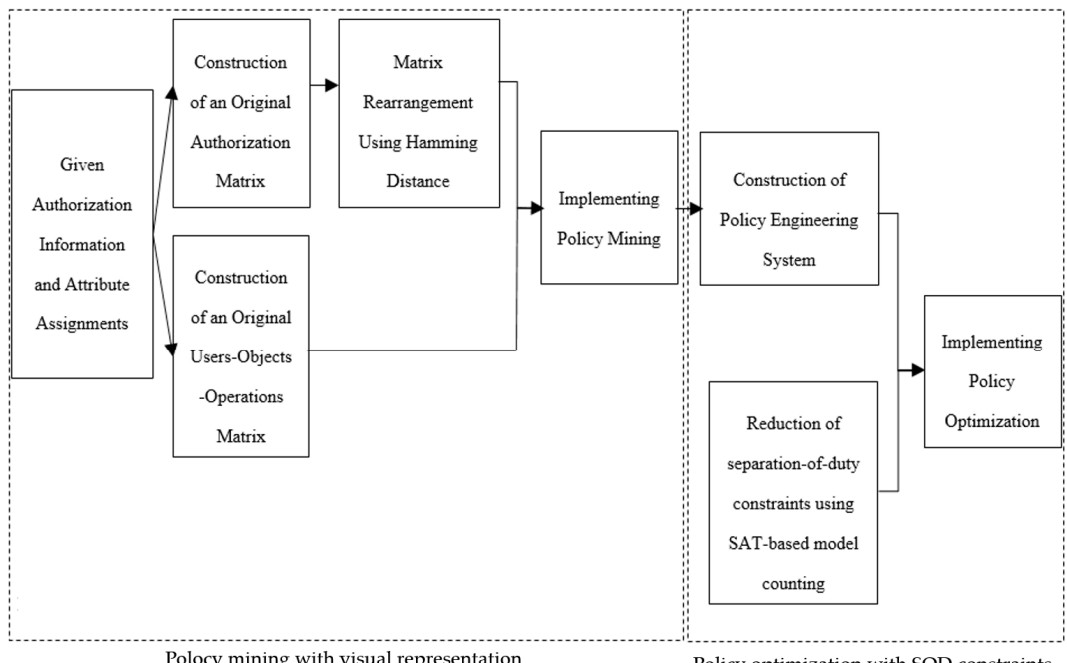

**Figure 2.** Flow chart of PEO_VR&SOD.

*4.1. Policy Mining with Visual Representation*

4.1.1. Preprocessing

First, we present the definitions of the matrix representations for *UUAV* and *OOAV*, respectively.

**Definition 1.** *(UUA) Represents the Boolean matrix form corresponding to a given UUAV. It can be represented as follows:*

$$UUA[i][j] = \begin{cases} 1, if \quad user \quad u_i \quad has \quad the \quad attribute \quad value \quad in \quad the \quad j^{th} column \\ 0, otherwise \end{cases}. \tag{5}$$

**Definition 2.** *(OOA) Represents the Boolean matrix form corresponding to a given OOAV. It can be represented as follows:*

$$OOA[i][j] = \begin{cases} 1, if \quad object \quad o_i \quad has \quad the \quad attribute \quad value \quad in \quad the \quad j^{th} column \\ 0, otherwise \end{cases}. \tag{6}$$

To satisfy basic policy mining, we construct an extensive users-objects-operations relationship for a given set of authorizations *A*. This relationship can be equivalently denoted as a constructed matrix $UOP^A$, which is a generalized Cartesian product of the matrix $UUA^A$ and the matrix $OOA^A$ for the requesting users and requested objects, respectively. The rows of $UOP^A$ correspond to all the possible user–object pairs, and the columns correspond to the attribute-value pairs of the users and objects, as well as the operations in *A*.

**Example 1.** *Given an authorization list A, a matrix $UUA^A$ of the attribute value assignments for users, and a matrix $OOA^A$ of the attribute value assignments for objects in* Tables 1–3, *respectively, then, the matrix $UOP^A$ can be constructed as in Table 4, where the notations uav and oav represent the attribute values of users and objects, respectively.*



**Table 1.** Authorization list $A$.

| User | Object | Operation |
|---|---|---|
| $u_1$ | $o_1$ | $op_1$ |
| $u_2$ | $o_1$ | $op_1$ |
| $u_2$ | $o_1$ | $op_2$ |
| $u_3$ | $o_2$ | $op_1$ |
| $u_3$ | $o_2$ | $op_2$ |
| $u_4$ | $o_2$ | $op_1$ |

**Table 2.** Matrix $UUA^A$.

| User | $uav_1$ | $uav_2$ | $uav_3$ | $uav_4$ |
|---|---|---|---|---|
| $u_1$ | 0 | 1 | 1 | 0 |
| $u_2$ | 1 | 0 | 1 | 0 |
| $u_3$ | 1 | 0 | 0 | 1 |
| $u_4$ | 0 | 1 | 0 | 1 |

**Table 3.** Matrix $OOA^A$.

| Object | $oav_1$ | $oav_2$ | $oav_3$ |
|---|---|---|---|
| $o_1$ | 1 | 0 | 1 |
| $o_2$ | 0 | 1 | 1 |

**Table 4.** Matrix $UOP^A$.

| User-Object | $uav_1$ | $uav_2$ | $uav_3$ | $uav_4$ | $oav_1$ | $oav_2$ | $oav_3$ | $op_1$ | $op_2$ |
|---|---|---|---|---|---|---|---|---|---|
| $u_1$-$o_1$ | 0 | 1 | 1 | 0 | 1 | 0 | 1 | 1 | 0 |
| $u_2$-$o_1$ | 1 | 0 | 1 | 0 | 1 | 0 | 1 | 1 | 1 |
| $u_3$-$o_1$ | 1 | 0 | 0 | 1 | 1 | 0 | 1 | 0 | 0 |
| $u_4$-$o_1$ | 0 | 1 | 0 | 1 | 1 | 0 | 1 | 0 | 0 |
| $u_1$-$o_2$ | 0 | 1 | 1 | 0 | 0 | 1 | 1 | 0 | 0 |
| $u_2$-$o_2$ | 1 | 0 | 1 | 0 | 0 | 1 | 1 | 0 | 0 |
| $u_3$-$o_2$ | 1 | 0 | 0 | 1 | 0 | 1 | 1 | 1 | 1 |
| $u_4$-$o_2$ | 0 | 1 | 0 | 1 | 0 | 1 | 1 | 1 | 0 |

To mine suitable authorization rules from the constructed matrix *UOP*, we need to find the attribute–value pairs for users and objects, such that the values of their corresponding cells in one row are 1, as is the value of the operation cell. Meanwhile, there is not any row where the value of the operation cell is 0 while retaining the values of the cells as 1 for the same attribute-value pairs. The various combinations of different attribute-value pairs that meet the requirements are referred to as the ABAC authorization rules, the number of which needs to be minimum.

In other words, our method involves finding the minimal set of rules from the *UOP* that can cover all the rows with the operation columns having values of 1. Meanwhile, the different attribute-value pairs in the set do not present in any row with the operation column containing a value of 0.

To find a suitable and minimal set of rules, we present two guiding principles as follows:

(1) Principle 1: Assume that $k_1$ and $k_2$ are two different combinations of attribute-value pairs included in rules $ar_1$ and $ar_2$, respectively. If $k_1 \subseteq k_2$ (that is, the number of attribute-value pairs in $k_2$ is greater than that in $k_1$), then the authorizations covered by $k_2$ can also be covered by $k_1$; moreover,

$k_2$ is more restricted than $k_1$. The authorizations covered by rule *ar* increase in number as the number of attribute-value pairs in *ar* decreases. Thus, we should choose short-length rules for the number of attribute-value pairs in any rule.

(2) Principle 2: To discover a minimal set of rules while ensuring a short length in any rule, we decompose the *UOP* into two submatrices that consist of operation columns with values of 1 and 0. We denote these columns as $UOP_{op=1}$ and $UOP_{op=0}$ and sort $UOP_{op=1}$ in ascending order according to the number of values of 1 in any row. We attempt to mine rules to cover all the authorizations corresponding to $UOP_{op=1}$.

However, the constructed *UOP* becomes much more confusing and complex as the number of user–object pairs increases. In other words, it is difficult to analyze and identify the authorization rules from such a representation. Therefore, we need to find an alternative for the list of authorizations to further facilitate policy mining while making the authorization representation more visually appealing and understandable.

### 4.1.2. Visual Representation for Authorizations

**Definition 3.** *($A_{op}$) Represents the Boolean matrix form for a list of authorizations A with operation op, where the rows correspond to users, and the columns correspond to objects. This can be represented as follows:*

$$A_{op}[i][j] = \begin{cases} 1, if \quad u_i \quad is \quad allowed \quad to \quad perform \quad op \quad on \quad o_j \\ 0, otherwise \end{cases}. \tag{7}$$

**Example 2.** *An illustrative ABAC authorization matrix $A_{op}$ is shown in Table 5, which includes 10 users and 10 objects. An equivalently rearranged matrix $A_{op}'$ is presented in Table 6. We only mark the cells whose values are 1 for an intuitive representation. Obviously, it is more convenient to analyze and handle $A_{op}'$, which provides a motivation for functionally representing $A_{op}$.*

**Table 5.** Original matrix $A_{op}$.

|  | $o_1$ | $o_2$ | $o_3$ | $o_4$ | $o_5$ | $o_6$ | $o_7$ | $o_8$ | $o_9$ | $o_{10}$ |
|---|---|---|---|---|---|---|---|---|---|---|
| $u_1$ | 1 | 1 |  |  |  |  |  |  |  | 1 |
| $u_2$ | 1 | 1 |  |  |  |  |  |  |  | 1 |
| $u_3$ |  |  | 1 | 1 |  |  | 1 | 1 |  |  |
| $u_4$ |  |  | 1 | 1 |  |  | 1 | 1 |  |  |
| $u_5$ | 1 |  |  |  |  | 1 |  |  |  |  |
| $u_6$ | 1 | 1 |  |  |  |  |  |  |  | 1 |
| $u_7$ | 1 |  |  |  |  | 1 |  |  |  |  |
| $u_8$ |  |  | 1 | 1 |  |  | 1 | 1 |  |  |
| $u_9$ |  |  | 1 | 1 |  |  | 1 | 1 |  |  |
| $u_{10}$ | 1 | 1 |  |  |  |  |  |  |  | 1 |

**Table 6.** Rearranged matrix $A_{op}'$.

|  | $o_5$ | $o_9$ | $o_3$ | $o_4$ | $o_7$ | $o_8$ | $o_6$ | $o_1$ | $o_2$ | $o_{10}$ |
|---|---|---|---|---|---|---|---|---|---|---|
| $u_8$ |  |  | 1 | 1 | 1 | 1 |  |  |  |  |
| $u_9$ |  |  | 1 | 1 | 1 | 1 |  |  |  |  |
| $u_3$ |  |  | 1 | 1 | 1 | 1 |  |  |  |  |
| $u_4$ |  |  | 1 | 1 | 1 | 1 |  |  |  |  |
| $u_5$ |  |  |  |  |  |  | 1 | 1 |  |  |
| $u_7$ |  |  |  |  |  |  | 1 | 1 |  |  |
| $u_1$ |  |  |  |  |  |  |  | 1 | 1 | 1 |
| $u_2$ |  |  |  |  |  |  |  | 1 | 1 | 1 |
| $u_6$ |  |  |  |  |  |  |  | 1 | 1 | 1 |
| $u_{10}$ |  |  |  |  |  |  |  | 1 | 1 | 1 |

To visually represent the authorization matrix, we use the Hamming distance to rearrange it as defined below.

**Definition 4.** *(Visual representation problem for authorizations) Given an authorization matrix $A_{op}$, and a Hamming distance list D between any two rows of $A_{op}$, find a rearranged matrix $A_{op}'$ such that: (1) The sum of distances between the adjacent rows of $A_{op}'$ is minimum, and (2) the submatrices $\{s_1, s_2, \dots\}$ are intuitively included in $A_{op}'$ while covering all the cells with values of 1, which can be formalized as follows:*

$$\begin{cases} \min(\sum_i Dis(A_{op}'[i], A_{op}'[i+1])), \forall Dis(A_{op}'[i], A_{op}'[i+1]) \in D \\ A_{op}'[i'][j'] = 1, \forall s_k \in \{s_1, s_2, \dots\} \quad included \quad in \quad A_{op}', \forall A_{op}'[i'][j'] \quad in \quad s_k \end{cases} . \tag{8}$$

According to Definition 5, we present the process of matrix rearrangement in Algorithm 1.

---

**Algorithm 1** Matrix rearrangement

---

**Input:** original matrix $A_{op}$
**Output:** rearranged matrix $A_{op}'$
1. Initialize $A_{op}' = A_{op}$;
2. Represent $A_{op}'$ as a list of row vectors: $A_{op}'[1], A_{op}'[2], \dots$;
3. Identify matrix $D_r$ of the Hamming distances between any two row vectors such that
$\forall i,j: D_r[i][j] = Dis(A_{op}'[i], A_{op}'[j])$;
4. **for** each $A_{op}'[i]$ in $A_{op}'$ **do**
5. **if** ($\exists A_{op}'[j]$: $D_r[i][j] < D_r[i][i+1]$) **then**
6.   $swap(A_{op}'[i+1], A_{op}'[j])$;
7. **end if**
8. **end for**

---

The distance matrix $D_r$ for the original matrix in Table 5 is shown in Table 7, where both the rows and columns correspond to the row vectors, and the values of cells are the Hamming distances between any two rows. According to Algorithm 1, the same (or similar) row vectors are clustered by choosing the minimal distances. Similarly, we can also rearrange $A_{op}'$ to cluster the same (or similar) column vectors. These clusters of rows and columns then form a submatrix, and several such submatrices together cover all the cells of $A_{op}$ where $A_{op}[i][j] = 1$. Specifically, since $D_r[10][1] = D_r[10][2] = D_r[10][6] = 0$, we can swap row vector $A_{op}'[1]$ with $A_{op}'[8]$, $A_{op}'[2]$ with $A_{op}'[9]$, and $A_{op}'[6]$ with $A_{op}'[7]$, respectively, and the result in step 1 is shown in Table 8, which clusters the same row vectors together, such as $\{A_{op}'[6], A_{op}'[1], A_{op}'[2], A_{op}'[10]\}$, $\{A_{op}'[5], A_{op}'[7]\}$, and $\{A_{op}'[8], A_{op}'[9], A_{op}'[3], A_{op}'[4]\}$. Subsequently, we implement the rearrangement towards the column vectors. Since $D_c[4][7] = D_c[4][8] = 0$, we swap the corresponding columns in step 2, and the result is shown in Table 9. Similarly, we swap the second column with the ninth column in step 3, and the result is shown in Table 10. According to the minimal distances between the adjacent columns, we swap the first column with the eighth column in step 4, and the result is shown in Table 11, which is consistent with the visual representation in Table 6.

**Table 7.** Distance matrix $D_r$.

|  | $A_{op}'[1]$ | $A_{op}'[2]$ | $A_{op}'[3]$ | $A_{op}'[4]$ | $A_{op}'[5]$ | $A_{op}'[6]$ | $A_{op}'[7]$ | $A_{op}'[8]$ | $A_{op}'[9]$ | $A_{op}'[10]$ |
|---|---|---|---|---|---|---|---|---|---|---|
| $A_{op}'[1]$ | 0 | 0 | 7 | 7 | 3 | 0 | 3 | 7 | 7 | 0 |
| $A_{op}'[2]$ | 0 | 0 | 7 | 7 | 3 | 0 | 3 | 7 | 7 | 0 |
| $A_{op}'[3]$ | 7 | 7 | 0 | 0 | 6 | 7 | 6 | 0 | 0 | 7 |
| $A_{op}'[4]$ | 7 | 7 | 0 | 0 | 6 | 7 | 6 | 0 | 0 | 7 |
| $A_{op}'[5]$ | 3 | 3 | 6 | 6 | 0 | 3 | 0 | 6 | 6 | 3 |
| $A_{op}'[6]$ | 0 | 0 | 7 | 7 | 3 | 0 | 3 | 7 | 7 | 0 |
| $A_{op}'[7]$ | 3 | 3 | 6 | 6 | 0 | 3 | 0 | 6 | 6 | 3 |
| $A_{op}'[8]$ | 7 | 7 | 0 | 0 | 6 | 7 | 6 | 0 | 0 | 7 |
| $A_{op}'[9]$ | 7 | 7 | 0 | 0 | 6 | 7 | 6 | 0 | 0 | 7 |
| $A_{op}'[10]$ | 0 | 0 | 7 | 7 | 3 | 0 | 3 | 7 | 7 | 0 |

**Table 8.** Step 1 for rearrangement.

|  | $o_1$ | $o_2$ | $o_3$ | $o_4$ | $o_5$ | $o_6$ | $o_7$ | $o_8$ | $o_9$ | $o_{10}$ |
|---|---|---|---|---|---|---|---|---|---|---|
| $u_8$ |  |  | 1 | 1 |  |  | 1 | 1 |  |  |
| $u_9$ |  |  | 1 | 1 |  |  | 1 | 1 |  |  |
| $u_3$ |  |  | 1 | 1 |  |  | 1 | 1 |  |  |
| $u_4$ |  |  | 1 | 1 |  |  | 1 | 1 |  |  |
| $u_5$ | 1 |  |  |  |  | 1 |  |  |  |  |
| $u_7$ | 1 |  |  |  |  | 1 |  |  |  |  |
| $u_6$ | 1 | 1 |  |  |  |  |  |  |  | 1 |
| $u_1$ | 1 | 1 |  |  |  |  |  |  |  | 1 |
| $u_2$ | 1 | 1 |  |  |  |  |  |  |  | 1 |
| $u_{10}$ | 1 | 1 |  |  |  |  |  |  |  | 1 |

**Table 9.** Step 2 for rearrangement.

|  | $o_1$ | $o_2$ | $o_3$ | $o_4$ | $o_8$ | $o_7$ | $o_6$ | $o_5$ | $o_9$ | $o_{10}$ |
|---|---|---|---|---|---|---|---|---|---|---|
| $u_8$ |  |  | 1 | 1 | 1 | 1 |  |  |  |  |
| $u_9$ |  |  | 1 | 1 | 1 | 1 |  |  |  |  |
| $u_3$ |  |  | 1 | 1 | 1 | 1 |  |  |  |  |
| $u_4$ |  |  | 1 | 1 | 1 | 1 |  |  |  |  |
| $u_5$ | 1 |  |  |  |  |  | 1 |  |  |  |
| $u_7$ | 1 |  |  |  |  |  | 1 |  |  |  |
| $u_6$ | 1 | 1 |  |  |  |  |  |  |  | 1 |
| $u_1$ | 1 | 1 |  |  |  |  |  |  |  | 1 |
| $u_2$ | 1 | 1 |  |  |  |  |  |  |  | 1 |
| $u_{10}$ | 1 | 1 |  |  |  |  |  |  |  | 1 |

**Table 10.** Step 3 for rearrangement.

|       | $o_1$ | $o_9$ | $o_3$ | $o_4$ | $o_7$ | $o_8$ | $o_6$ | $o_5$ | $o_2$ | $o_{10}$ |
|-------|-------|-------|-------|-------|-------|-------|-------|-------|-------|----------|
| $u_8$ |       |       | 1 | 1 | 1 | 1 |   |   |   |   |
| $u_9$ |       |       | 1 | 1 | 1 | 1 |   |   |   |   |
| $u_3$ |       |       | 1 | 1 | 1 | 1 |   |   |   |   |
| $u_4$ |       |       | 1 | 1 | 1 | 1 |   |   |   |   |
| $u_5$ | 1 |   |   |   |   |   | 1 |   |   |   |
| $u_7$ | 1 |   |   |   |   |   | 1 |   |   |   |
| $u_6$ | 1 |   |   |   |   |   |   |   | 1 | 1 |
| $u_1$ | 1 |   |   |   |   |   |   |   | 1 | 1 |
| $u_2$ | 1 |   |   |   |   |   |   |   | 1 | 1 |
| $u_{10}$ | 1 |   |   |   |   |   |   |   | 1 | 1 |

**Table 11.** Step 4 for rearrangement.

|       | $o_5$ | $o_9$ | $o_3$ | $o_4$ | $o_8$ | $o_7$ | $o_6$ | $o_1$ | $o_2$ | $o_{10}$ |
|-------|-------|-------|-------|-------|-------|-------|-------|-------|-------|----------|
| $u_8$ |   |   | 1 | 1 | 1 | 1 |   |   |   |   |
| $u_9$ |   |   | 1 | 1 | 1 | 1 |   |   |   |   |
| $u_3$ |   |   | 1 | 1 | 1 | 1 |   |   |   |   |
| $u_4$ |   |   | 1 | 1 | 1 | 1 |   |   |   |   |
| $u_5$ |   |   |   |   |   |   | 1 | 1 |   |   |
| $u_7$ |   |   |   |   |   |   | 1 | 1 |   |   |
| $u_6$ |   |   |   |   |   |   |   | 1 | 1 | 1 |
| $u_1$ |   |   |   |   |   |   |   | 1 | 1 | 1 |
| $u_2$ |   |   |   |   |   |   |   | 1 | 1 | 1 |
| $u_{10}$ |   |   |   |   |   |   |   | 1 | 1 | 1 |

### 4.1.3. Policy Mining

In this subsection, we take the rearranged authorization matrix $A_{op}'$ and the matrices *UUA* and *OOA* as inputs and present the process of policy mining in Algorithm 2.

In Algorithm 2, we first partition the rearranged matrix into *k* small matrices according to the number of submatrices in Lines 1–3, Line 4 defines three sets of rules, *Initial_rules*, *Uninitial_rules*, and *Candidate_rules*, and initializes them. From Line 5, we start to mine the rules in each submatrix. For each submatrix $A_{op}'_i$, we construct $UOP^{A_{op}'_i}$ and decompose it into $UOP^{A_{op}'_i}_{op=1}$ and $UOP^{A_{op}'_i}_{op=0}$ according to Principle 2 in Line 6. We insert all the attribute-value pairs present in $UOP^{A_{op}'_i}_{op=1}$ into the set *Initial_rules* and insert all the attribute-value pairs present in $UOP^{A_{op}'_i}_{op=0}$ into the set *Uninitial_rules* in Lines 7–8. Next, in Lines 9–21, we use double loops for the sets *Candidate_rules* and *Initial_rules* to choose the short-length rules that are not present in *Uninitial_rules* and consider them as candidate rules. To further simplify the rules in set *Candidate_rules*, we estimate whether the length of any rule (that is, the number of attribute-value pairs) can be reduced in the last few lines. As shown in Table 11, three submatrices (<{$u_8$,$u_9$,$u_3$,$u_4$}, {$o_3$,$o_4$,$o_7$,$o_8$}>, <{$u_5$,$u_7$}, {$o_6$,$o_1$}>, and <{$u_1$,$u_2$,$u_6$,$u_{10}$}, {$o_1$,$o_2$,$o_{10}$}>), which separate the rearranged matrix into three partitions, are visually appealing. It is more convenient and feasible to derive rules from each small partition, and the detailed mining processes are omitted owing to the limited space.

---

**Algorithm 2** Policy mining

---

**Input:** rearranged matrix $A_{op}'$, matrices $UUA$ and $OOA$

**Output:** set $AR$ of authorization rules

1. Identify the number of visual submatrices in $A_{op}'$ as $k$;

2. Based on the visual submatrices, separate $A_{op}'$ into $n$ partitions: $A_{op}'_1$, $A_{op}'_2$, $\ldots$ , and $A_{op}'_k$. In each partition, the columns correspond to the same set of objects, and the rows correspond to different sets of users;

3. According to $k$ sets of users in different partitions, separate $UUA$ into $k$ partitions: $UUA_1$, $UUA_2$, $\ldots$ , and $UUA_n$;

4. Define and initialize the sets of rules: *Initial_rules* = ø, *Uninitial_rules* = ø, *Candidate_rules* = ø;

5.**for** each $A_{op}'_i$ in $\{A_{op}'_1, A_{op}'_2, \ldots , A_{op}'_k\}$ **do**

6. Construct matrix $UOP^{A_{op}'_i}$ using the Cartesian product of the $UUA_i$ and $OOA$, and decompose it into $UOP^{A_{op}'_i}_{op=1}$ and $UOP^{A_{op}'_i}_{op=0}$;

7. Identify combinations of the different attribute-value pairs present in all rows of $UOP^{A_{op}'_i}_{op=1}$ and sort them in ascending order according to the number of attribute-value pairs. Consider them as initial rules and insert them into *Initial_rules*;

8. Identify the combinations of different attribute-value pairs present in all rows of $UOP^{A_{op}'_i}_{op=0}$. Do not consider them as rules and insert them into *Uninitial_rules*;

9. **for** each combination of attribute-value pairs *ar* in *Candidate_rules* **do**

10.    **for** each combination of attribute-value pairs *ar'* in *Initial_rules* **do**

11.     **if** (*ar* is not null)$\wedge$(*ar* $\subseteq$ *ar'*) **then**

12.      **continue**;/*authorizations covered by *ar'* has been covered by *ar*\*/

13.     **else**

14.      **if** (*ar*$\cap$*ar'*) $\notin$ *Uninitial_rules* **then**/*authorizations covered by both *ar* and *ar'* are allowed*/

15.       *Candidate_rules* = (*Candidate_rules*\\{*ar*})$\cap$\{*ar*$\cap$*ar'*\};

16.     **else**        /*authorizations covered by *ar'* are only allowed*/

17.       *Candidate_rules* = *Candidate_rules*$\cap$\{*ar'*\};

18.     **end if**

19.     **end if**

20.    **end for**

21. **end for**

22. **for** each rule *car* in *Candidate_rules* **do**

23. **for** each attribute-value pair *a* in *car* **do**

24.    **if** (*car*\\{*a*}) $\notin$ *Uninitial_rules* **then**

25.     *car* = *car*\\{*a*};

26.    **end if**

27. **end for**

28. *AR* = *AR*$\cup$\{*car*\};

29. **end for**

30.**end for**

---

Computational complexity: Choosing suitable rules mainly depends on the double loops in Lines 9–21 and the estimation operations in Lines 22–29. Assume that the number of submatrices is $k$ and that the number of rows with operation columns containing values of 1 in each submatrix $A_{op}'_i$ is $x$; then, the total execution time of the algorithm is $O(k \times (x^2 \times (|A_{op}'_I - x|) + x^2))$, which is influenced by the number of partitions and the size of each submatrix.

*4.2. Policy Optimization with Separation-of-Duty Constraints*

First, we present three status functions among users, authorization rules, and access tuples of the SOD constraint, defined as follows.

**Definition 5.** *(Status functions)*

*(1)* *user_rules$_\gamma$(u) represents a set of authorization rules that allows user u to perform operations on objects under the system status γ;*

*(2)* *tuple_rules$_\gamma$(t) represents a set of authorization rules that allows access to tuple t for a given SOD constraint under the system status γ;*

*(3)* *user_tuples$_\gamma$(u) represents a set of tuples of a given SOD constraint that is authorized to user u under the system status γ. It can be formalized as*

$$user\_tuples_\gamma(u) = \left\{ t \middle| \exists ar \in user\_rules_\gamma(u) : ar \in tuple\_rules_\gamma(t) \right\}. \tag{9}$$

To demonstrate whether the SOD constraints can be satisfied, the different status functions in Definition 6 are used in the following example.

**Example 3.** *Given a constructed ABAC system status γ, where the rules associated with each user are shown in Table 12. Consider a set of SOD constraints ω = {sod$_1$, sod$_2$}, where sod$_1$ = <{t$_1$, t$_2$, t$_3$}, 2> and sod$_2$ = <{t$_4$, t$_5$, t$_6$}, 3>. The rules that are allowed to access any tuple with respect to sod$_1$ and sod$_2$ are presented in Tables 13 and 14, respectively.*

**Table 12.** Sets of different user_rules$_\gamma$(u).

| User | Rules |
|------|-------|
| $u_1$ | $\{ar_1, ar_2, ar_4, ar_5\}$ |
| $u_2$ | $\{ar_3, ar_7\}$ |
| $u_3$ | $\{ar_1, ar_2, ar_4, ar_5\}$ |
| $u_4$ | $\{ar_6\}$ |

**Table 13.** Sets of different tuple_rules$_\gamma$(t) with respect to $sod_1$.

| Tuple | Rules |
|-------|-------|
| $t_1$ | $\{ar_7\}$ |
| $t_2$ | $\{ar_3\}$ |
| $t_3$ | $\{ar_1, ar_2\}$ |

**Table 14.** Sets of different tuple_rules$_\gamma$(t) with respect to $sod_2$.

| Tuple | Rules |
|-------|-------|
| $t_4$ | $\{ar_4, ar_5, ar_6\}$ |
| $t_5$ | $\{ar_3\}$ |
| $t_6$ | $\{ar_1, ar_2\}$ |

According to $sod_1$ and Table 13, at least two users are required to collaborate and together have $t_1$, $t_2$, and $t_3$, and all these access tuples can be done using either the rules $ar_1$, $ar_3$, and $ar_7$ or the rules $ar_2$, $ar_3$, and $ar_7$. In other words, no user can own these three rules at the same time. Table 8 shows that $sod_1$ can be satisfied under $\gamma$. Similarly, according to $sod_2$ and Table 14, any two users cannot own rules $ar_1$, $ar_3$, and $ar_5$ at the same time. Table 12 demonstrates, however, that users $u_1$ and $u_2$ can access all the tuples of $sod_2$ through these three rules. Hence, $sod_2$ cannot be satisfied under $\gamma$. Therefore, immediately enforcing SOD constraints in ABAC systems appears to be intractable.

Reducing the problem of verifying SOD constraints in an ABAC system into the construction of MEAR constraints includes two phases: (1) Converting *k-n* SOD constraints into *k-n* SOAR constraints and (2) constructing *t-m* MEAR constraints from *k-n* SOAR constraints.

### 4.2.1. Construction of k-n SOAR Constraints from k-n SOD Constraints

Given a constructed rule set *AR* and a specific SOD constraint $<\{t_1, t_2, \ldots, t_n\}, k>$ under system status $\gamma$, we convert the input instance $(<\{t_1, t_2, \ldots, t_n\}, k>, AR)$ into an intermediate form according to the following steps:

**Step 1.** For each access tuple $t_i$ in the constraint, we identify set *tuple_rules*$_\gamma(t_i)$ as a substitute for $t_i$;

**Step 2.** Replace rule set *AR* with a union of different set of rules associated with each tuple in the constraint.

Then, the corresponding intermediate form, which is denoted as *CIF*, can be represented as *CI F* $=<\{S_1, S_2, \ldots S_n\}, AR'>$, where $S_i = tuple\_rules_\gamma(t_i)$, and $AR' = \bigcup_{i=1}^{n} S_i \subseteq AR$.

According to the conversion, two conclusions can be made:

(1) The process of the conversion does not take value $k$ of the constraint into consideration because, for each $k'$-$n$ SOAR constraint constructed, $k'$ takes the value of $k$ in the following process of constructions.

(2) $AR'$ in *CIF* does not contain the rules in $(AR \backslash AR')$ because the rules not in $AR'$ are not relevant to the construction of the SOAR constraints.

A SAT solver takes the formula in CNF as the input parameter for solving the problem, which can identify the total number of different types of such assignments. For the given intermediate form $CIF = <\{S_1, S_2, \ldots S_n\}, AR'>$, we next construct a CNF formula $F$ corresponding to *CIF* through the following steps:

**Step 1.** For each rule $ar_i$ in $AR'$, create the corresponding literal variable $\overline{ar_i}$;

**Step 2.** For each $S_i$ and $AR'$, create the corresponding clause using the disjunction of different literals, which is formalized as $\overline{ar_1} \vee \overline{ar_2} \vee \ldots$;

**Step 3.** Construct a CNF formula using the conjunction of different clauses, which is represented as $F = (\bigwedge_{i=1}^{n} S_i clause) \wedge AR' clause$.

Thus, a set of several different literals that can satisfy a truth assignment for $F$ forms the SOAR constraint, and all sets of such literals form different types of SOAR constraints, where the value of $k$ remains constant.

**Example 4.** *Consider a SOD constraint sod* $= <\{t_1, t_2, t_3, t_4\}, 2>$ *in the ABAC system status $\gamma$. The corresponding rules associated with each tuple are given in Table 15.*

**Table 15.** Sets of different *tuple_rules(t)* for *sod*.

| Tuple | Rules |
|---|---|
| $t_1$ | $\{ar_4\}$ |
| $t_2$ | $\{ar_2\}$ |
| $t_3$ | $\{ar_1, ar_3\}$ |
| $t_4$ | $\{ar_2, ar_3, ar_4\}$ |

First, we convert *sod* into the intermediate form $CIF = <\{\{ar_4\}, \{ar_2\}, \{ar_1, ar_3\}, \{ar_2, ar_3, ar_4\}\}, \{ar_1, ar_2, ar_3, ar_4\}>$. Next, we construct the Boolean formula $F$ for *CIF* as $F = \overline{ar_4} \wedge \overline{ar_2} \wedge (\overline{ar_1} \vee \overline{ar_3}) \wedge (\overline{ar_2} \vee \overline{ar_3} \vee \overline{ar_4}) \wedge (\overline{ar_1} \vee \overline{ar_2} \vee \overline{ar_3} \vee \overline{ar_4})$. It is readily verified that $\{\overline{ar_1}, \overline{ar_2}, \overline{ar_4}\}$ can satisfy a truth assignment for F, and $<\{ar_1, ar_2, ar_4\}, 2>$ is regarded as a SOAR constraint. Moreover, $\{\overline{ar_2}, \overline{ar_3}, \overline{ar_4}\}, \overline{ar_2}, , \overline{ar_3}, \overline{ar_4}\}$ can also satisfy such assignments, and $<\{ar_2, ar_3, ar_4\}, 2>$, $<\{ar_1, ar_2, ar_3, ar_4\}, 2>$ are the other two SOAR constraints corresponding to *sod*.

### 4.2.2. Construction of t-m MEAR Constraints from k-n SOAR Constraints

In this subsection, we determine how to construct MEAR constraints for the given SOAR constraints. Using the notions of the *k-n* SOAR constraint and the *t-m* MEAR constraint, as well as the status function $user\_rules_\gamma(u)$ in Definition 6, we can define the safety and satisfiability of the system status as the following.

**Definition 6.** *(Safety of the system status, $safe_{soar}(\gamma)$) Given an ABAC system status $\gamma$ and a k-n SOAR constraint soar = <{$ar_1, ar_2, \ldots ar_n$}, k>, if any set of (k − −1) users cannot have all the n rules under $\gamma$, then $\gamma$ is safe with respect to soar, which is denoted as $safe_{soar}(\gamma)$ = 1. Otherwise, $\gamma$ is unsafe with respect to soar, which is denoted as $safe_{soar}(\gamma)$ = 0. Let the set of different k-n SOAR constraints be $\xi$ = {$soar_1, soar_2, \ldots$ }; if $\gamma$ is safe with respect to each $soar_i$, then $\gamma$ is safe with respect to $\xi$, which is denoted as $safe_\xi(\gamma)$ = 1. Otherwise, $\gamma$ is unsafe with respect to $\xi$, which is denoted as $safe_\xi(\gamma)$ = 0.*

The safety of the system status is formally expressed as:

$$\forall\{u_1, u_2, \ldots, u_{k-1}\} \subset U : \{ar_1, ar_2, \ldots, ar_n\} \overset{k-1}{\underset{i=1}{\cup}} user\_rules_\gamma(u_i) \Rightarrow safe_{soar}(\gamma) = 1. \tag{10}$$

The unsafety of the system status is formally expressed as:

$$\exists\{u_1, u_2, \ldots, u_{k-1}\} \subset U : \overset{k-1}{\underset{i=1}{\cup}} user\_rules_\gamma(u_i) \supseteq \{ar_1, ar_2, \ldots, ar_n\} \Rightarrow safe_{soar}(\gamma) = 0. \tag{11}$$

**Definition 7.** *(Satisfiability of the system status, $satisfied_{mear}(\gamma)$) Given an ABAC system status $\gamma$ and a t-m MEAR constraint mear = <{$ar_1, ar_2, \ldots ar_m$}, t>, if no user is allowed to have t or more of these m rules under $\gamma$, then $\gamma$ is satisfied with respect to mear, which is denoted as $satisfied_{mear}(\gamma)$ = 1. Otherwise, $\gamma$ is unsatisfied with respect to mear, which is denoted as $satisfied_{mear}(\gamma)$ = 0. Let the set of different t-m MEAR constraints be $\psi$ = {$mear_1, mear_2, \ldots$ }; if $\gamma$ is satisfied with respect to each $mear_i$, then $\gamma$ is satisfied with respect to $\psi$, which is denoted as $satisfied_\psi(\gamma)$ = 1. Otherwise, $\gamma$ is unsatisfied with respect to $\psi$, which is denoted as $satisfied_\psi(\gamma)$ = 0.*

The satisfiability of the system status is formally expressed as:

$$\forall u \in U : \left|user\_rules_\gamma(u) \cap \{ar_1, ar_2, \ldots ar_m\}\right| < t \Rightarrow satisfied_{mear}(\gamma) = 1. \tag{12}$$

The unsatisfiability of the system status is formally expressed as:

$$\exists u \in U : \left|user\_rules_\gamma(u) \cap \{ar_1, ar_2, \ldots ar_m\}\right| \geq t \Rightarrow satisfied_{mear}(\gamma) = 0. \tag{13}$$

**Statement 1.** *Given an ABAC system status $\gamma$ and a set of SOAR constraints $\xi$ = {$soar_1, soar_2, \ldots$ }, the process for verifying whether $\gamma$ is safe with respect to $\xi$ is in P.*

**Proof.** We prove this statement through the following three steps: (1) Use status function $user\_rules_\gamma()$ and identify $user\_rules_\gamma(u)$ for some user $u$ under $\gamma$ (2). Based on the formal expression in Definition 7, identify the number of the rules in $user\_rules_\gamma(u)$ present in any *sod*, which is represented as *s*, and (3) compare *s* with *k* included in *sod*. These steps can be readily computed. If $|U|,|AR|$, and $|\xi|$ are used to represent the total number of users, the number of authorization rules and the number of SOAR constraints under $\gamma$, respectively, then the computational complexity for verifying whether $safe_\xi(\gamma)$ is true is $O(|U|\times|AR|\times|\xi|)$, which is polynomial time. Thus, the verification process for enforcing the SOAR constraints is in *P*. □

**Statement 2.** *Given an ABAC system status $\gamma$ and a set $\psi$ of MEAR constraints, the process of verifying whether $\gamma$ is satisfied with respect to $\psi$ is also in P.*

**Proof .** This verification is similar to that of Theorem 1, so the detailed process is omitted due to limited space. Here, the enforcement of MEAR constraints under $\gamma$ is also available in polynomial time. □

**Definition 8.** *(Implicit enforcement of SOAR constraints) Given a k-n SOAR constraint soar and a t-m MEAR constraint set* $\psi = \{mear_1, mear_2, \dots \}$ *in the system status* $\gamma$, *soar can be implicitly enforced by* $\psi$ *if and only if* $\forall mear \in \psi : satisfied_{mear}(\gamma) \Rightarrow safe_{soar}(\gamma).$

Then, we present an approach for constructing *t-m* MEAR constraints in Algorithm 3, which takes a *k-n* SOAR constraint constructed from a *k-n* SOD constraint as input and outputs a set of MEAR constraints that can implicitly enforce SOAR.

---

**Algorithm 3** Construction of *t-m* MEAR constraints

---

**Input:** *k-n* SOAR constraint *soar* = <$\{ar_1, ar_2, \dots ar_n\}, k$>, where $2 \leq k \leq n$
**Output:** set $\psi$ of *t-m* MEAR constraints
1. Initialize $\psi = \emptyset$;
2. **if** $k == 2$ **then**
3. $\quad \psi = \{<\{ar_1, ar_2, \dots ar_n\}, n>\}$;
4. **else if** $k == n$ **then**
5. $\quad \psi = \{<\{ar_1, ar_2, \dots ar_n\}, 2>\}$;
6. **else**
7. $\quad$ **for** $t = 2$ to $\left\lfloor \frac{n-1}{k-1} \right\rfloor + 1$ **do**
8. $\quad\quad m = (k\text{-}1) \times (t\text{-}1)+1$;
9. $\quad\quad$ **for** any subset $\{ar_1, ar_2, \dots ar_m'\}$ in $\{ar_1, ar_2, \dots ar_n\}$ **do**
10. $\quad\quad\quad \psi = \psi \cup \{<\{ar_1, ar_2, \dots ar_m'\}, t>\}$;
11. $\quad\quad$ **end for**
12. $\quad$ **end for**
13. **end if**

---

As observed from Lines 2–5 in Algorithm 3, two lemmas can be determined with the following:

**Lemma 1.** *Given a 2-n SOAR constraint, it can be efficiently enforced by the t-m MEAR constraint if and only if m = n, and t = n.*

**Lemma 2.** *Given a n-n SOAR constraint, it can be efficiently enforced by the t-m MEAR constraint if and only if m = n, and t = 2.*

**Theorem 1.** Given a k-n SOAR constraint (k > 2), it can be efficiently enforced by the constructed t-m MEAR constraints through Algorithm 3 if $t \leq \left( \left\lfloor \frac{n-1}{k-1} \right\rfloor + 1 \right)$

**Proof .** According to Definitions 7 and 8, any user is allowed to have $(t-1)$ rules at most, and any $(k-1)$ users are allowed to have $(k-1) \times (t-1)$ rules at most. We use the method of contradiction to prove this theorem. Without a loss of generality, assume that $t = \left\lfloor \frac{n-1}{k-1} \right\rfloor + 1 + 1$; the *k-n* SOAR constraint is still satisfied, i.e., the number of rules associated with $(k-1)$ users is $(k-1) \times \left( \left\lfloor \frac{n-1}{k-1} \right\rfloor + 2 - 1 \right) \approx n - 1 + k - 1 > n$, which breaches the *k-n* SOAR constraint. Thus, the assumption is false. □

To further demonstrate the effectiveness of Algorithm 3, we next provide an example to show the construction of MEAR constraints from a given SOD constraint.

**Example 5.** *Given a set of authorization rules AR = $\{ar_1, ar_2, ar_3, ar_4, ar_5\}$ and a SOD constraint sod = <$\{t_1, t_2, t_3, t_4, t_5\}, 3$> in the ABAC system status* $\gamma$, *let the sets of tuple\_rules$_\gamma$(t) with respect to sod be $\{ar_1, ar_2\}$, $\{ar_3\}$, $\{ar_2, ar_3\}$, $\{ar_4\}$, and $\{ar_4, ar_5\}$, respectively.*

Then, for *sod*, the corresponding *soars* that can be constructed using the method of SAT-based model counting are as follows:

$\{<\{ar_2, ar_3, ar_4\}, 3>, <\{ar_1, ar_3, ar_4\}, 3>, <\{ar_1, ar_2, ar_3, ar_4\}, 3>, <\{ar_1, ar_3, ar_4, ar_5\}, 3>, <\{ar_2, ar_3, ar_4, ar_5\}, 3>, <\{ar_1, ar_2, ar_3, ar_4, ar_5\}, 3>\}$.

The different sets of *mears* constraints corresponding to each *soar* according to Algorithm 3 are given in Table 16.

**Table 16.** Different sets of *mears* with respect to *soars*.

| Soar | Mears |
|---|---|
| $<\{ar_2, ar_3, ar_4\}, 3>$ | $\{<\{ar_1, ar_2, ar_4\}, 2>\}$ |
| $<\{ar_1, ar_3, ar_4\}, 3>$ | $\{<\{ar_2, ar_3, ar_4\}, 2>\}$ |
| $<\{ar_1, ar_2, ar_3, ar_4\}, 3>$ | $\{<\{ar_1, ar_2, ar_3\}, 2>, <\{ar_2, ar_3, ar_4\}, 2>, <\{ar_1, ar_2, ar_4\}, 2>, <\{ar_1, ar_3, ar_4\}, 2>\}$ |
| $<\{ar_1, ar_3, ar_4, ar_5\}, 3>$ | $\{<\{ar_1, ar_3, ar_4\}, 2>, <\{ar_3, ar_4, ar_5\}, 2>, <\{ar_1, ar_3, ar_5\}, 2>, <\{ar_1, ar_4, ar_5\}, 2>\}$ |
| $<\{ar_2, ar_3, ar_4, ar_5\}, 3>$ | $\{<\{ar_2, ar_3, ar_4\}, 2>, <\{ar_3, ar_4, ar_5\}, 2>, <\{ar_2, ar_3, ar_5\}, 2>, <\{ar_2, ar_4, ar_5\}, 2>\}$ |
| $<\{ar_1, ar_2, ar_3, ar_4, ar_5\}, 3>$ | $\{<\{ar_1, ar_2, ar_3\}, 2>, <\{ar_1, ar_2, ar_4\}, 2>, <\{ar_1, ar_2, ar_5\}, 2>, <\{ar_1, ar_3, ar_4\}, 2>, <\{ar_1, ar_3, ar_5\}, 2>, <\{ar_1, ar_4, ar_5\}, 2>, <\{ar_2, ar_3, ar_4\}, 2>, <\{ar_2, ar_3, ar_5\}, 2>, <\{ar_2, ar_4, ar_5\}, 2>, <\{ar_3, ar_4, ar_5\}, 2>, <\{ar_1, ar_2, ar_3, ar_4, ar_5\}, 3>\}$ |

## 5. Experimental Evaluations

To evaluate the efficiency and effectiveness of the PEO_VR&SOD, we next implemented experiments using both real and synthetic datasets and compare the performance of PEO_VR&SOD with that of existing methods. All the experiments were carried out on a standard desktop PC with an Intel i5–7400 CPU, 4 GB RAM, and a 160 GB hard disk running a 64-bit Windows 7 operating system. All simulations were compiled and run in Eclipse IDE under the Java Developer environment.

### 5.1. Performance Comparison with the Xu-Stoller and VisMAP in Real Datasets

First, we consider the following real datasets from [10], as shown in Table 17. These datasets have been widely used for research on different methods of ABAC policy mining, such as Xu-Stoller and VisMAP. The first seven columns in the table represent the corresponding dataset name, numbers of users, user attributes, objects, object attributes, number of all possible attribute values for users and objects, and number of authorizations. To evaluate the efficiency of our method in the policy mining stage, we consider the number of authorization rules in the policy and the execution time as evaluation measures.

**Table 17.** Descriptions of real datasets.

| Dataset | $|U|$ | $|UA|$ | $|O|$ | $|OA|$ | $|Val|$ | $|A|$ | Xu-Stoller | | VisMAP | | Our Method | |
|---|---|---|---|---|---|---|---|---|---|---|---|---|
| | | | | | | | $|P|$ | T(s) | $|P|$ | T(s) | $|P|$ | T(s) |
| University | 20 | 6 | 34 | 5 | 76 | 168 | 10 | 0.02 | 10 | 0.02 | 10 | 0.02 |
| Healthcare | 21 | 6 | 16 | 7 | 55 | 51 | 11 | 0.02 | 7 | 0.02 | 7 | 0.02 |
| Project Management | 16 | 7 | 40 | 6 | 77 | 189 | 19 | 0.03 | 12 | 0.03 | 12 | 0.03 |

We repeatedly implement the experiments 5 times in the above three datasets, take an average of overall values, and compare the results including $|P|$ and T with the performance of Xu-Stoller and VisMAP in the last columns of Table 17. The number of authorization rules mined using our method is less than that using Xu-Stoller and is equal to that of VisMAP. Meanwhile, in the table, there is almost no difference in execution time for all three methods. This is because there is a small number of users, objects, and authorizations in any real dataset; thus, it is feasible to find suitable rules using Xu-Stoller, though both our method (PEO_VR&SOD) and VisMAP adopt visual representation technology for a given authorization relationship before policy mining. Thus, PEO_VR&SOD performs as well as Xu-Stoller and VisMAP on the small University, Healthcare, and Project Management datasets.

### *5.2. Performance Comparison with VisMAP in Synthetic Datasets*

Since it is very difficult to find suitable real datasets, we next construct synthetic datasets with specific parameters, where the number of users varies from 100 to 1000 with a step of 100; the number of objects are 100, 200, 500, and 1000; and the attributes for users and objects randomly take values from the above real datasets. To evaluate the efficiency of PEO_VR&SOD, we take into consideration the number of mining rules and execution time as measures and compare the results with the performance of VisMAP. Additionally, as indicated in Algorithms 1 and 2, we partition the rearranged matrix by visual representations for the original matrix, while VisMAP directly separates an original matrix before rearrangement. If the synthetic datasets constructed are already sufficiently visual, then the authorization matrices need to be neither rearranged nor partitioned. Therefore, we first consider the following cases with no partition.

We repeatedly implement the experiments 10 times in different synthetic datasets and take the average value. The results are shown in Figures 3 and 4. For Figure 3, the lateral axis represents the number of users, and the vertical axis represents the number of rules in the policy. For Figure 4, the lateral axis represents the number of users, and the vertical axis represents the comparison of execution time.

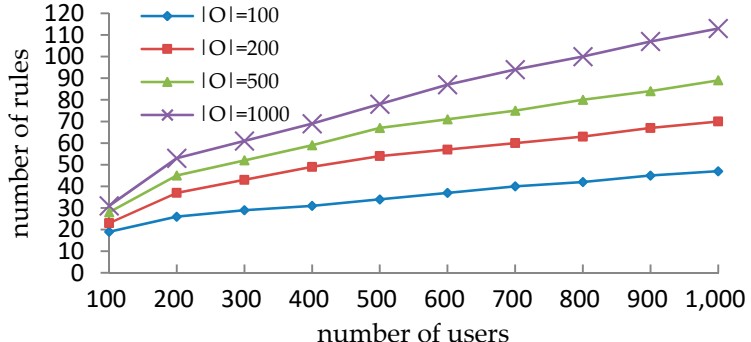

**Figure 3.** Comparison of rules for different numbers of users and objects.

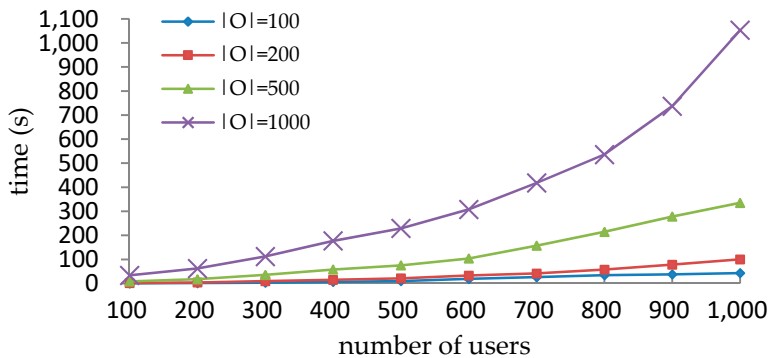

**Figure 4.** Comparison of time for different numbers of users and objects.

Figure 3 shows that the number of rules varies with a varying number of users for different numbers of objects. If the number of objects is fixed at 100, and the number of users varies from 100 to 1000, then the number of rules varies from 19 to 47, which increases slightly with an increase in the number of users; however, if the number of objects is fixed at 1000, the number of rules varies from 31 to 113, showing a clear increase. Conversely, if we fix the number of users at 100 and 1000, then the number of rules also increases with an increasing number of both objects and users. Obviously, the varying number of mining rules tends to grow linearly, particularly when the number of objects is small. This is because, the greater the number of users is, the more rules that can be constructed from the mining process.

Figure 4 shows that the execution time varies with a varying number of users for different numbers of objects. If the number of objects is less than 500, then the execution time is always below 335 s and tends to grow linearly. However, if the number is greater than 500, then the execution time increases exponentially to around 1100 s with 1000 users and 1000 objects, which is unacceptable. That is attributed to the fact that the dimension of an authorization matrix becomes larger and larger with an increasing number of users and objects, and a large-size matrix requires more time to be rearranged using Algorithm 1.

To demonstrate the interpretability of the mining process, we employ synthetic datasets on which VisMAP has been executed, where the numbers of users and objects are 100, 200, 500, and 1000. In addition, the number of partitions for each rearranged matrix is 1, 2, and 4. We repeatedly implement the experiments 10 times on these synthetic datasets, output the number of rules and the execution time, and take their average values. Comparisons of the results for PEO_VR&SOD and VisMAP are shown in Table 18.

**Table 18.** Performance comparison with VisMAP.

| |U| | |O| | |Partitions| | | | | | | | | | | | |
|---|---|---|---|---|---|---|---|---|---|---|---|---|---|
| | | 1 | | 2 | | 4 | | 1 | | 2 | | 4 | |
| | | VisMAP | | | | | | PEO_VR&SOD | | | | | |
| | | |P| | T(s) | |P| | T(s) | |P| | T(s) | |P| | T(s) | |P| | T(s) | |P| | T(s) |
| 100 | 100 | 19 | 0.59 | 38 | 0.72 | 62 | 0.35 | 19 | 0.59 | 22 | 0.84 | 30 | 0.93 |
| 200 | 100 | 26 | 1.77 | 55 | 1.11 | 84 | 1.03 | 26 | 1.77 | 31 | 1.98 | 42 | 2.14 |
| 500 | 100 | 34 | 10.18 | 76 | 5.38 | 108 | 5.9 | 34 | 10.18 | 42 | 10.73 | 53 | 11.95 |
| 1000 | 100 | 47 | 43.28 | 96 | 20.46 | 197 | 10.92 | 47 | 43.28 | 56 | 43.61 | 74 | 44.17 |
| 100 | 200 | 23 | 1.63 | 48 | 1.57 | 78 | 0.95 | 23 | 1.63 | 26 | 1.87 | 34 | 2.02 |
| 200 | 200 | 37 | 4.25 | 77 | 3.33 | 114 | 2.14 | 37 | 4.25 | 42 | 5.01 | 53 | 5.61 |
| 500 | 200 | 54 | 21.54 | 96 | 12.36 | 146 | 10.78 | 54 | 21.54 | 56 | 22.34 | 62 | 23.24 |
| 1000 | 200 | 70 | 100.4 | 145 | 43.01 | 197 | 39.77 | 70 | 100.4 | 81 | 105.34 | 98 | 106.11 |
| 100 | 500 | 28 | 8.27 | 50 | 9.78 | 89 | 4.72 | 28 | 8.27 | 33 | 8.77 | 43 | 9.07 |
| 200 | 500 | 45 | 17.64 | 91 | 16.41 | 156 | 9.39 | 45 | 17.64 | 50 | 18.13 | 65 | 18.55 |
| 500 | 500 | 67 | 75.23 | 141 | 49.92 | 224 | 33.57 | 67 | 75.23 | 74 | 76.02 | 87 | 76.93 |
| 1000 | 500 | 89 | 334.98 | 207 | 163.96 | 304 | 115.50 | 89 | 334.98 | 99 | 335.44 | 115 | 336.01 |
| 100 | 1000 | 31 | 33.84 | 60 | 47.46 | 109 | 10.09 | 31 | 33.84 | 35 | 34.11 | 47 | 34.89 |
| 200 | 1000 | 53 | 62.47 | 100 | 70.63 | 180 | 33.43 | 53 | 62.47 | 58 | 62.88 | 68 | 63.56 |
| 500 | 1000 | 78 | 248.87 | 175 | 174.30 | 282 | 100.21 | 78 | 248.87 | 87 | 249.19 | 103 | 250.17 |
| 1000 | 1000 | 113 | 1052.78 | 218 | 695.67 | 335 | 452.52 | 113 | 1052.78 | 126 | 1053.11 | 155 | 1053.88 |

Table 18 shows that, for both methods, as the number of users and objects increases, the number of rules increases slightly, while the execution time increases more clearly whenever |Partitions| is equal to 1, 2, or 4, which is consistent with the experimental analyses shown in Figures 3 and 4. Second, as the number of partitions increases, the number of rules increases clearly, while the execution time decreases in VisMAP; however, both the number of rules and the execution time increase slightly in PEO_VR&SOD. Specifically, when the number of users and objects is fixed at 1000 and 100, respectively, for the former, the number of rules varies from 47 to 197 with an increase of more than 300%, while the time varies from 43.28 to 10.92 with a decrease of more than 75%; for the latter, the number of rules increases from 47 to 74, and the time increases from 43.28 to 44.17. This is attributed to the fact that the original matrix in VisMAP is separated into several partitions before policy mining and spends little time on the small datasets. However, the rearranged matrix is separated during mining and spends more time with the number of partitions according to Algorithm 2. Although VisMAP outperforms PEO_VR&SOD for execution time, the number of rules using our proposal is less than that of VisMAP, and the overhead in time for PEO_VR&SOD is minimal compared to the benefit of the number of the mining rules.

In the above evaluations, we implement the experiments for VisMAP and PEO_VR&SOD with the same experimental setups: the number of users and objects varies while keeping the number of their attributes and attribute values constant. However, there are other properties, such as the length and size of authorization rules, need to be considered for evaluating the performance of Xu-Stoller. Furthermore, no partition exists in Xu-Stoller. Therefore, Xu–Stoller will not outperform VisMAP or PEO_VR&SOD for execution time on the same datasets.

*5.3. Performance Comparison with Xu-Stoller on Synthetic Datasets*

To further evaluate the efficiency of our method, we construct synthetic datasets and consider a similar setup of parameters to that used in Xu-Stoller. These parameters involve the number of users (|U|), the number of objects (|O|), the number of attribute-value pairs of users (|UAV|), the number of attribute-value pairs of objects (|OAV|), the maximum number of rules used to construct the datasets (|RC|), and the maximum length of any mining rule (|RL|). In addition, to fairly and effectively compare the performance of PEO_VR&SOD and Xu–Stoller, we convert our constructed datasets into data formats in which Xu–Stoller can be executed. We assume that each constructed authorization includes a single permission because any rule in our policy engineering system involves only one operation.

To study the effect of different parameters on the mining results, we consider three scenarios: (1) |UAV| and |OAV| vary from 20 to 40 with a step of 5, (2) |RC| varies from 20 to 50 with a step of 5, and (3) |RL| varies from 2 to 5 with a step of 1. One parameter varies while keeping the others constant when any scenario happens. Since the effect of variations in users or objects is considered (as shown in Figures 3 and 4), the values of |U| and |O| are fixed at 1000 and 100, respectively, in the following. The descriptions of the datasets and their parameters are shown in the first six columns in Table 19. We repeatedly implement the experiments 10 times on different datasets and take the median value. The number of the mining rules for the proposed algorithm ($|P_{PEO\_VR\&SOD}|$) and that of Xu-Stoller ($|P_{Xu\text{-}Stoller}|$) are presented in the last two columns of the table. Table 19 shows that, for all these datasets, both PEO_VR&SOD and Xu–Stoller discover almost the same number of rules, which is less than the maximum number of rules used to construct the datasets.

Moreover, we compare the average execution time with that of Xu–Stoller in Figures 5–7 with error bars, where the lateral axis represents the varying values of different parameters, and the vertical axis represents changes in execution time.

**Table 19.** Comparison of rules with Xu-Stoller for different parameters.

| $|U|$ | $|O|$ | $|UAV|$ | $|OAV|$ | $|RC_{max}|$ | $|RL_{max}|$ | $|P_{Xu\text{-}Stoller}|$ | $|P_{PEO\_VR\&SOD}|$ |
|------|------|--------|--------|-----------|-----------|-----------------|-----------------|
| 1000 | 100 | 20 | 20 | 30 | 5 | 27 | 26.31 |
| 1000 | 100 | 25 | 25 | 30 | 5 | 25.67 | 25.67 |
| 1000 | 100 | 30 | 30 | 30 | 5 | 26.67 | 25.11 |
| 1000 | 100 | 35 | 35 | 30 | 5 | 26.67 | 25.37 |
| 1000 | 100 | 40 | 40 | 30 | 5 | 27.53 | 26.89 |
| 1000 | 100 | 25 | 25 | 20 | 5 | 17.67 | 17.53 |
| 1000 | 100 | 25 | 25 | 40 | 5 | 36 | 35.31 |
| 1000 | 100 | 25 | 25 | 50 | 5 | 43 | 42.83 |
| 1000 | 100 | 25 | 25 | 30 | 4 | 25.67 | 25.39 |
| 1000 | 100 | 25 | 25 | 30 | 3 | 24.67 | 24.64 |
| 1000 | 100 | 25 | 25 | 30 | 2 | 24 | 24 |

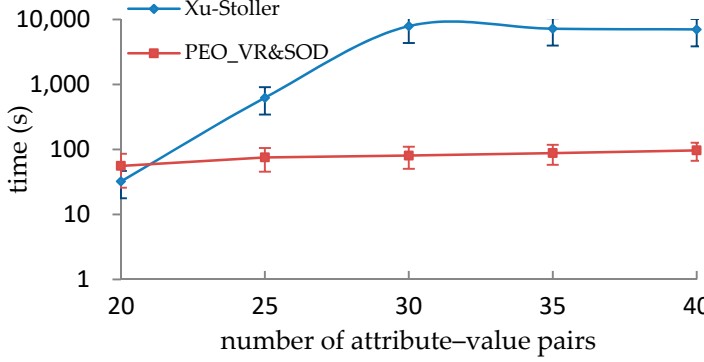

**Figure 5.** Comparison of time with Xu-Stoller for different numbers of attribute-value pairs.

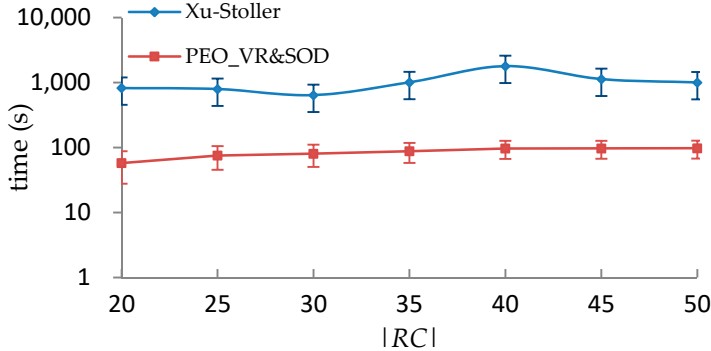

**Figure 6.** Comparison of time with Xu-Stoller for different $|RC|$ values.

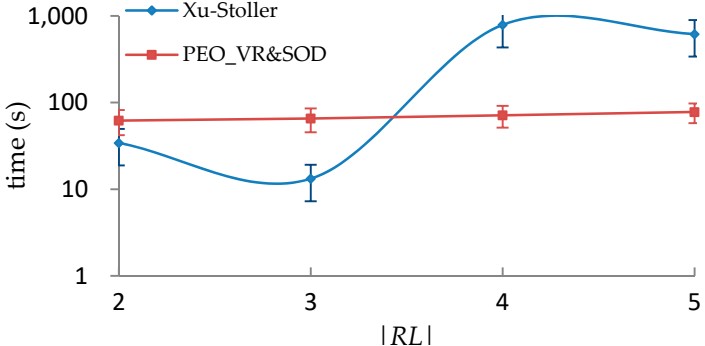

**Figure 7.** Comparison of time with Xu-Stoller for different $|RL|$ values.

These figures show that, for PEO_VR&SOD, the execution time tends to grow linearly and does not obviously vary as the number of attributes values, the value of |*RC*|, and/or the value of |*RL*| vary. However, for Xu–Stoller, Figure 5 demonstrates that the execution time first increases exponentially and then decreases gradually with an increasing number of attribute values. Specifically, the time increases remarkably from around 33 s to around 8000 s when the number of attribute values varies from 20 to 30 and then decreases gradually to around 7000 s when the value is close to 40. Further, the time required in our method is below 100 s for any case, while Xu-Stoller requires significantly more time with an increasing number of attribute values. PEO_VR&SOD runs faster than Xu-Stoller because the former eliminates the redundant attributes from each candidate rule, as shown in Lines 22–29 of Algorithm 2, while the latter takes more time in the generalization step while eliminating attribute expressions with constraints. Figures 6 and 7 demonstrate that the execution time irregularly fluctuates up and down with increasing values of |*RC*| and |*RL*|. Detailed analyses are not discussed in this paper, similar to the analyses in [27]. Thus, PEO_VR&SOD outperforms Xu–Stoller on these datasets.

In the above evaluations, we implement the experiments for Xu-Stoller and PEO_VR&SOD with the same experimental setups: considering not only the varying number of users and objects as well as that of their attributes, but also number of rules used to construct the datasets and the maximum length of any mining rule. However, we do not need to consider all of these setups for VisMAP on the same datasets. Therefore, VisMAP will outperform Xu–Stoller as well as PEO_VR&SOD for execution time on the same datasets.

### 5.4. Performance Evaluation of Enforcement of SOD Constraints

To evaluate the effectiveness of PEO_VR&SOD, we study the performance of enforcing SOD constraints in the policy optimization stage, which can be converted into a study on the effects of enforcing MEAR constraints. We also employed real datasets (as shown in Table 17) that were used to construct an ABAC system in the policy mining stage. Further, the SOD constraints were synthetically constructed similar to the examples in this paper, such as the 3–5 SOD constraints. Moreover, the Rel-SAT model counter [37] is used for constructing the SOAR constraints. As shown by Definition 8 and Statement 2, the verification of the MEAR constraints is primarily affected by the number of users, as well as that of the rules. Therefore, we implemented experiments in datasets where the number of users varies from 20 to 50 with a step of 10, and the number of rules varies from 5 to 20 with a step of 5. The execution time for verification with a varying number of users and rules is shown in Figure 8.

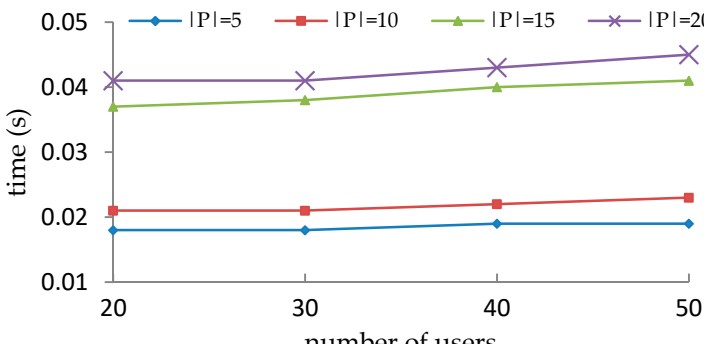

**Figure 8.** Execution time for different numbers of users and rules.

Figure 8 shows that the execution time tends to grow linearly and does not obviously vary with an increasing number of users for each specific policy configuration. However, when the number of users remains constant, the time varies clearly with a varying number of rules. Specifically, for 10 rules, the time always remains around 0.02 s as the number of users varies, while for 30 users, the time varies from around 0.02 to 0.04 as the number of rules varies. This occurs because the more rules

there are, the larger the size of the set of SOAR constraints constructed by the Rel-SAT model counter will be, and the greater the number of MEAR constraints that will be generated using Algorithm 3. Obviously, the total time for the verification of MEAR constraints increases with an increasing number of constraints.

*5.5. Discussions*

From the above analyses, we present some discussions in the following:

(1) For the sake of brevity, we do not consider the environments, such as time and locations. However, consideration of time factor in the constrained policy engineering is also an interesting topic. Take the dynamic SOD for example, the *k-n* dynamic SOD can be modified as *dsod* < {$t_1, t_2, \ldots, t_i \ldots, t_n$}, *k*, [*bt*, *et*]>, where [*bt*, *et*] is a time interval. It states that the enforcement of the SOD constraint is valid from *bt* to *et*. Similarly, both the authorization set and rule set also need to be modified with time intervals.

(2) Figures 3 and 4 aim to demonstrate the performance of our method with different number of users and objects, and the result is consistent with the analysis of the computational complexity of Algorithm 2. In fact, VisMAP and Xu–Stoller have the similar performance, which have been presented in the state-of-the-art literatures and also can be observed from Tables 18 and 19. Thus, we do not make the similar figures for VisMAP and Xu-Stoller in our work.

(3) Figures 5–7 demonstrate variations of execution time for Xu-Stoller with different number of attribute-value pairs of users and objects, different maximum number of rules used to construct the datasets, and different maximum length of any mining rule, respectively. However, these attribute properties are constant for the evaluation of VisMAP, and thus we do not make the same type of graphs for VisMAP.

(4) Although PEO_VR&SOD performs better for mining optimal sets of rules in the above real and synthetic datasets, Either VisMAP or Xu-Stoller does well for some scenarios, and the corresponding synthetic datasets are more appropriate for efficiency evaluations of VisMAP or Xu-Stoller. Specifically, it is observed from Table 18 that, VisMAP outperforms PEO_VR&SOD for execution time with the increasing number of partitions. From Figure 7, the number of users and objects are fixed at 1000 and 100, respectively. Xu–Stoller also outperforms PEO_VR&SOD when the maximum length of any mining rule takes varies of 2 and 3.

## 6. Conclusions

A novel method for policy-engineering optimization called PEO_VR&SOD was proposed in this paper. We first used the visual technique with Hamming distance to reduce the policy-mining scale and presented the policy mining algorithm. Then, we used the method of SAT-based model counting to convert the SOD constraints into the corresponding SOAR constraints and constructed MEAR constraints to implicitly enforce the SOAR constraints in the constructed policy-engineering system. As a result, the proposed method can successfully address the stated problems of enhancing interpretability while mining a minimal set of rules and implicitly enforcing SOD constraints in a constructed ABAC system. The experiments demonstrated that the proposed method is efficient and effective. However, a few interesting issues remain to be solved. One issue is how to implement PEO_VR&SOD in systems such as blockchains, wireless sensor networks, and the internet of things. Another issue is how to implement cardinality constraints for policy-engineering optimizations in future work.

**Author Contributions:** Conceptualization, W.S.; methodology, W.S.; validation, W.S. and H.S.; formal analysis, W.S.; data curation, H.S. and H.X.; writing—original draft preparation, W.S. All authors have read and agreed to the published version of the manuscript.

**Funding:** This work was partially supported by the Natural Science Foundation of China (61501393), the Natural Science Foundation of Henan Province of China (182300410145, 182102210132), and Foundation of Henan Educational Committee under Contract No. 20B20031.

**Conflicts of Interest:** The authors declare no conflict of interest.

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
