# Peer review of "Policy-Engineering Optimization with Visual Representation and Separation-of-Duty Constraints in Attribute-Based Access Control"

_futureinternet, doi:10.3390/fi12100164_

Round 1

Reviewer 1 Report

The paper presents an innovative method of policy-engineering in ABAC using visual representation and SoD constraints. The authors have written a good paper, which is easy to read. The state-of-the-art review is quite comprehensive and progresses logically as well as the mathematical apparatus is quite adequate. There are not many issues with the paper except some minor ones as detailed below: - authors argue that for the sake of brevity they do not consider environments (e.g. dynamic SoD) However, it would be worthwhile for authors to discuss the impact of this assumption. What would change, or would need to be changed in your framework to accomodate dynamic SoD ? - Figure 1 - it is not clear what single and double arrow heads represent - Algorithm 2 represents visual policy mining, however there a visual representation of this step might have been of interest to the readers (similar to what you put for algorithm 1 - matrix rearrangement) - you repeat your experiments 5 or 10 times. Why? Is it enough for meaningful results ? - for results in table 13 you take the average values, while for results in Fig 2 and 3 you take median values. Can you clarify why ? - please re-check the figure and axis labels in fig. 2 and 3, they are confusing. I suppose for Fig. 2 vertical axis you mean "number of rules in the policy" and for Fig. 3 label you mean "comparison of execution time" - for results in Table 14 you state that you "repeatedly" implemented the experiments, but you do not mention how many times you repeated the experiments - Results in Table 14 seem to be better for VisMAP or PEO_VR&SOD depending on the scenarion (number of users and objects). However, you bold only the results in PEO_VR&SOD. It would have been more visually clear if you bolded the better result either in VisMAP or PEO_VR&SOD. - It is arguable to say that VisMAP is outperformed by PEO_VR&SOD since it is clear from the table that sometimed VisMAP outperforms PEO_VR&SOD. Rather, it would be better to discuss scenarios where it would be better to use VisMAP vs scenarios where it would be better to use PEO_VR&SOD. - The same can be said for Xu-Stoller. - In sec 5.3 you mention you construct synthetic datasets again. Are they the same datasets from Sec 5.2 ? If not, why was there a need for different datasets ? Could you not have used the same datasets ? - Fig 4-6 show a variation of (execution?) time for Xu-Stoller. Why did you not make the same type of graphs for VisMAP ? - Why did you not make Fig. 2 and 3 for VisMAP and Xu-Stoller ? - This reviewer suggests to make the experiment section more consistent (same types of graphs/tables) and also insert a subsection called discussion (can be the last subsection of Sec. 5) where you may discuss everything that was mentioned before: What would change, or would need to be changed in your framework to accomodate dynamic SoD; discuss scenarios where it would be better to use VisMAP or Xu-Stoller vs scenarios where it would be better to use PEO_VR&SOD; considerations on how to implement PEO_VR&SOD in blockchains, WSN or IoT etc.)

Author Response

Thanks for your helpful comments, and we have revised the paper. Please see the attachment.

Reviewer 2 Report

The authors are dealing with the concept of Attribute-Based Access Control (ABAC) and they propose a methodology for policy engineering optimization with visual representation and separation of duties constraints, which they call PEO_VR&SOD. The authors evaluated their proposal with real and synthetic datasets and compared it in terms of the number of generated rules in the policy and execution time.

The general impression of the submission is that it is full of equations and formulas without explaining how these equations contribute to the realization of the rules. It is difficult to read the manuscript and hard to follow the arguments of the authors. Thus, I suggest a thorough revision in order to improve the comprehensibility of the manuscript.

A background section that will present the fundamentals of ABAC will be useful. Currently, a reader, which is not familiar with the concept of ABAC, struggles to follow the rationale. Since ABAC is compared with RBAC throughout the paper, a comparison of their key characteristics is required (in order to understand the benefits of ABAC).

The purpose of the related work is to justify the novelty of their work, not just list what other researchers have done on the topic. The authors have to explain how their own work differentiates from the existing and advances the field.

Before the presentation of the technical details of their proposal (in section 4), it will be useful to present an abstract high-level description of what they are trying to achieve and how they achieve it. It is not evident how the building blocks of the proposal, such as the visual technique with Hamming distance or the rearranged of the authorization matrix contribute to policy engineering.

Also, it will be useful to explain how their proposal will benefit the administrators that desire to deploy ABAC on their system. Which are the requirements on their side and the practical implications?

The notion of “visual” is not evident.

The “Example 1” on the line 295 does not facilitate the discussion. I struggle to understand the relevance and connection of the example with the tables 1-4.

Please explain the benefits of the rearranged matrix.

The performance evaluation requires reorganization and re-execution. I do not see the reasons why the authors compare their proposal separately for each related work utilizing a different (synthetic) dataset (see section 5.2 comparison with VisMAP and section 5.2 comparison with Xu-Stoller). It is not obvious why they utilize a different synthetic dataset (with different attributes) to compare with each methodology. In my humble opinion, it will be more fair to compare their proposal for each synthetic dataset (with the different attributes) with both related works (both VisMAP and Xu-Stoller).

Furthermore, for the case of real datasets (section 5.1), I understand that all three methodologies (VisMAP, Xu-Stoller, and their own) exhibit the same performance. I am wondering whether these real datasets are also realistic, meaning that can exist in an actual environment or in the modern environments exist a great number of users, attributes, objects, etc., and thus the need for the synthetic dataset.

Regarding section 5.2, it is not obvious how their proposal outperforms VisMAP in terms of execution time. Obviously, in terms of time execution, their proposal requires more time. The authors simply justify that “This variation of time in our method is not obvious and can be accepted”, however it is not obvious why this can be accepted.

A required correction:

  1. 92: some preliminary results --> some preliminaries

Author Response

(The authors gave the same response as above.)

Round 2

Reviewer 1 Report

The authors have addressed many of my comments.

However, the discussion section should be revised. As it is written right now it is more of a justification of the limitation of their work and indication of future work (which should not be present in a journal paper - a conference paper is a work in progress which mandates a future work section, not a journal paper which should present a complete work) rather than a scientific discussion (e.g. "Next, we will improve the algorithms of PEO_VR&SOD or VisMAP, and search more desired datasets for implementing experiments in order to obtain more visually clear results." and "Currently, we do not simultaneously compare these three methods for each synthetic datasets, which is another limitation in our work").

Reviewer 2 Report

I will like to thank the authors for their effort to address the reviewer’s comments. However, there are still some issues that need clarification.

The response to comment 4 is not satisfactory. No high-level description of the proposal is provided. In the corresponding section, the authors should explain descriptive in a high-level manner the notion of their proposal.

The response to comment 6 is not satisfactory. Just modifying the term “visual” from some terms does not provide any explanation of how “visual” is the proposal.

Regarding comment 8, they should explain the steps that lead to the next rearranged matrix (Table 8-11). Furthermore, how is table 8 derived, and what does initially represent?

Regarding comment 9 (performance evaluation), I still can perceive the reasons why they do not evaluate each synthetic dataset with all 3 comparing proposals. If it is not feasible to evaluate all the synthetic datasets with all the 3 proposals, at least explain how the third (not evaluated) proposal will behave on each dataset.

Regarding comment 10, what the reviewer means is: in real environments (organizations, companies, etc) how many different users, attributes, objects exist. If the small datasets are taken from real indicative working environments, then why the need for the synthetic dataset?

Regarding comment 11, at least they should comment that the overhead in time for the proposal is minimal compared to the benefit of the number of the rules. It is not clear and not commented on what actually happens here.

Some required corrections:

  1. 37: A request --> a request
  2. 103: preliminary --> preliminaries

Round 3

Reviewer 2 Report

The manuscript is appropriate for publication in Future Internet journal.